# Increasing the Efficiency of Foundry Production by Changing the Technology of Pretreatment with Quartzite

Viktor Alekseevich Kukartsev [1], Aleksandr Ivanovich Cherepanov [1], Vladislav Viktorovich Kukartsev [2,3], Vadim Sergeevich Tynchenko [4,5,6,*], Sergei Olegovich Kurashkin [4,6,7,*], Roman Borisovich Sergienko [8], Valeriya Valerievna Tynchenko [9,10] and Kirill Aleksandrovich Bashmur [5]

[1] Department of Materials Science and Materials Processing Technology, Polytechnical Institute, Siberian Federal University, 660041 Krasnoyarsk, Russia; vkukarstev@sfu-kras.ru (V.A.K.); acherepanov@sfu-kras.ru (A.I.C.)

[2] Department of Informatics, Institute of Space and Information Technologies, Siberian Federal University, 660041 Krasnoyarsk, Russia; vlad_saa_2000@mail.ru

[3] Department of Information Economic Systems, Institute of Engineering and Economics, Reshetnev Siberian State University of Science and Technology, 660037 Krasnoyarsk, Russia

[4] Department of Information Control Systems, Institute of Computer Science and Telecommunications, Reshetnev Siberian State University of Science and Technology, 660037 Krasnoyarsk, Russia

[5] Department of Technological Machines and Equipment of Oil and Gas Complex, School of Petroleum and Natural Gas Engineering, Siberian Federal University, 660041 Krasnoyarsk, Russia; bashmur@bk.ru

[6] Digital Material Science: New Materials and Technologies, Bauman Moscow State Technical University, 105005 Moscow, Russia

[7] Laboratory of Biofuel Compositions, Siberian Federal University, 660041 Krasnoyarsk, Russia

[8] Machine Learning Department, Gini Gmbh, 80339 Munich, Germany; roman@gini.net

[9] Department of Computer Science, Institute of Space and Information Technologies, Siberian Federal University, 660041 Krasnoyarsk, Russia; 051301@mail.ru

[10] Department of Computer Science and Computer Engineering, Institute of Computer Science and Telecommunications, Reshetnev Siberian State University of Science and Technology, 660037 Krasnoyarsk, Russia

* Correspondence: vadimond@mail.ru (V.S.T.); scorpion_ser@mail.ru (S.O.K.); Tel.: +7-95-0973-0264 (V.S.T.)

**Abstract:** The efficiency of the production of foundry products depends on the reliable operation of the melting furnace including, therefore, the durability of its lining. The most common material adopted for the production of an acid furnace crucible lining is quartzite, in which during the pretreatment (heating to 800 °C followed by holding), a tridymite phase appears that maintains a constant volume at 840–1470 °C for a long time and provides high lining durability of 300–350 melts, but only when using melting temperature regimes not exceeding 1500 °C. However, the absence of iron scrap leads to the smelting of synthetic iron from only one steel scrap using higher melting temperatures (1550–1570 °C), which sharply reduces the lifetime of the lining to 220 melts. This work is devoted to research aimed at establishing technology for the pretreatment with the original quartzite, which ensures the formation of a phase state that successfully withstands elevated temperatures for a long time. The studies were carried out using a Bruker D8 ADVANCE diffractometer and a Shimadzu XRF-1800 X-ray wave-dispersive spectrometer. The work consisted of drying samples of the original quartzite at temperatures of 200 and 800 °C with subsequent exposure to temperatures of 200, 400, 600, 870, 1000, 1200, 1470 and 1550 °C. As a result, the conditions for pretreatment of quartzite were established, during which during its further use, a cristobalite phase can be obtained, which makes it possible manufacture a high-temperature lining that ensures its high durability. The introduction of this technology will ensure the efficiency of the production of foundry products for enterprises operating induction crucible furnaces at industrial frequency.

**Keywords:** clean technologies; cristobalite; efficiency; foundry; fuse; induction oven; lining; quartzite; manufacturing innovation; tridymite

## 1. Introduction

The development of a market economy that promotes economic growth leads to increased competition with suppliers of similar products to the market. For this reason, in the metallurgical industry, increasing importance is given to the reliable condition and efficient use of the main equipment. This foremost applies to melting furnaces, the correct operation of which should ensure the intensive reproduction of the main production assets [1,2]. This can only be achieved through timely and high-quality maintenance and repair, the cost of which is 8–12%. The technical and economic state of the enterprise itself depends on the solutions to this problem. The first issue is the labor productivity of workers, which is directly related to the technical condition and operability of the equipment and its downtime due to repairs. The main unit of the induction melting furnace is the crucible, composed of a refractory material that wears out over time, under the action of high temperatures [3]. Its replacement leads to the shutdown of the furnace (for a one-ton induction furnace, this procedure takes 1.5–2 days), which leads to downtime and reduces the efficiency of using the melting furnace itself. The most common material for the manufacture of a furnace crucible lining is quartzite, which, subject to its manufacturing technology and operating rules, provides resistance to 300–350 heat cycles.

Quartzite is a rock composed of quartz grains cemented together mainly by silica. In addition, it contains impurities, the main source of which is associated waste rock, in which quartz and quartzites occur. They are complex substances with a wide variety of properties, and the differences in these properties are determined by differences in the chemical composition and differences in the mutual arrangement of atoms (structure) [4]. The real characteristics of these structures with decoding into crystalline phases can only be established using diffraction research methods (X-ray, neutron diffraction or electron diffraction) [5,6]. Most of the impurities are present on the surfaces of pieces of quartz and quartzite in the form of "spreads" and calcium-containing crusts. To remove them, pretreatment is used, which consists of grinding quartzite and washing the clay spreads with water.

Quartz finds widespread application in industry. It is primarily utilized in the metallurgical industry, where it is used to obtain metallic silicon and its alloys, and in the manufacture of refractory acid-resistant material (dinas). In addition, due to the widespread use of induction melting furnaces, quartzite is actively used as the basis for the acid lining of their crucibles [7–11]. This lining has the highest resistance in the smelting of synthetic irons, as well as a low cost. It is also an indispensable material in the glass and optical industries and is beginning to be actively used for the manufacture of solar cells [12–16].

The properties of the quartzite used (of a specific ore deposit) depend first on its mineralogical, chemical and elemental composition. The main method of quality control for quartzite for the production of refractories and unshaped materials is the determination of the elemental composition, as well as the type and ratio of the crystalline phases (phase analysis). This control is carried out, as a rule, at all stages of production, from the analysis of the incoming raw materials to the evaluation of the final product. Different types of quartzites differ in terms of their mineral composition and content of impurity elements (especially manganese, iron, aluminum, titanium, boron and phosphorus), the presence and concentration of which determine the industrial use of the quartzite and the possibility of further enrichment [17–23]. Today, in Russia, for the lining of induction melting furnaces, quartzites from two deposits are used, and their impurity contents according to quarries are indicated in Table 1.

**Table 1.** Chemical composition of quartzites for lining induction furnaces.

| Field | Content, % | | | | | | | | | | |
|-------|------|-----------|-----|-----|------|-----------|----------|-----|-----|-------------------|------------------|
|       | $SiO_2$ | $Al_2O_3$ | CaO | MgO | $TiO_2$ | $Fe_2O_3$ | $P_2O_5$ | MnO | CaO | $Na_2O$ | $K_2O$ |
| Bobrovskoe | 97 | 1.1 | 0.6 | 0.05 | 0.06 | 0.6 | 0.01 | - | 0.16 | - | 0.021 |
| Pervouralskoe | 99 | 0.5 | 0.01 | 0.02 | 0.09 | 0.15 | 0.015 | 0.1 | 0.2 | 0.06 | 0.06 |

Developing technology for the preliminary preparation of raw materials should consider the later processes applied in the manufacturing of the final product intended for operation at a certain temperature. The main purpose of pretreatment with the original quartzite is to determine the conditions for obtaining phases of tridymite and cristobalite during its heating. The most studied process of using these phases is observed in the production of dinas, a refractory material for lining various furnace designs in metallurgy and foundry. Figure 1 shows the most common melting furnaces that require lining materials based on quartzite (electric arc and induction).

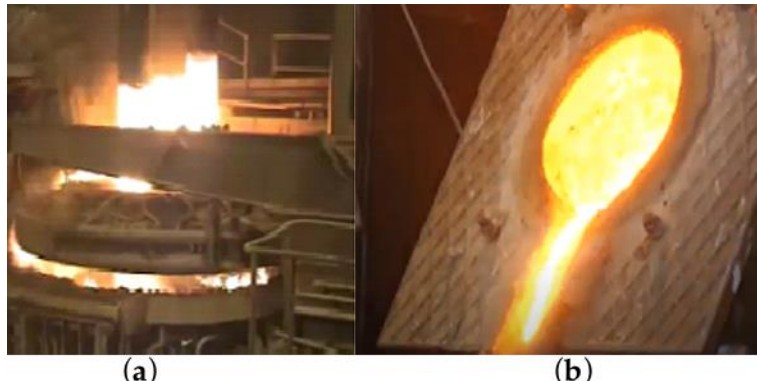

**Figure 1.** Melting furnaces: (**a**) electric arc furnace; (**b**) crucible induction furnace.

As practice has shown, dinas acquires the best properties, in which the main amount of quartz is converted into tridymite, which ensures its fire resistance at 1680–1730 °C. This dinas typically contains 50 to 70% tridymite and 20 to 30% cristobalite. During the manufacturing process, dinas is fired at a temperature of 1430 °C, which ensures the degeneration of quartz mainly into tridymite. When using higher temperatures, the intensive transformation of quartz into cristobalite occurs, which leads to a deterioration in the properties of products (the durability of the lining decreases due to the impact of sudden temperature changes during the operation of heating or electric arc furnaces). It was found that there are three zones of quartzite degeneration [24], shown in Figure 2.

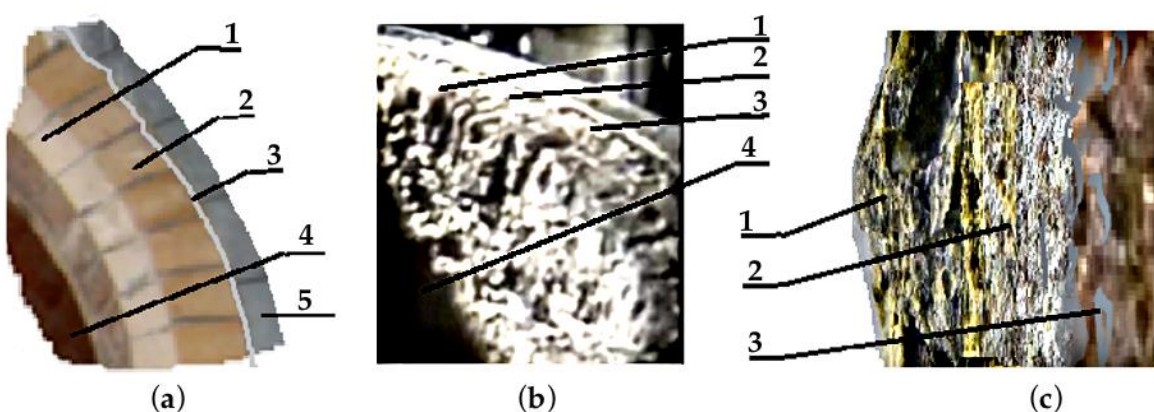

**Figure 2.** Dinas brick electric arc furnace lining and phase states of quartzite during operation: (**a**) new furnace lining: 1—dinas brick, 2—chamotte brick, 3—asbestos, 4—furnace bottom, 5—casing; (**b**) old lining: 1—cristobalite layer, 2—tridymite layer, 3—almost unchanged brick layer, 4—furnace bottom; (**c**) phase composition of the used brick: 1—cristobalite with a thin vitrified layer, 2—tridymite layer, 3—almost unchanged brick layer.

V. V. Lapin found that in the unregenerate (unchanged) zone of dinas brick, the content of residual quartzite was 11%. The cristobalite zone contained 81.6% cristobalite and 18.4% glass [25]. On this basis, he concluded that the tridymite zone can often be absent, since due to high temperatures, a dense, milky white cristobalite zone is immediately formed. It

prevents the normal course of physical and chemical processes that cause the formation of a tridymite zone.

The use of quartzite as part of the lining of industrial frequency induction melting furnaces intended for smelting synthetic iron is complicated by the following factors: frequent changes in temperature conditions, the ability to remove the lining without damaging the inductor after the end of its service life and resistance to intense mixing of the melt; these are shown in Figure 3c [26].

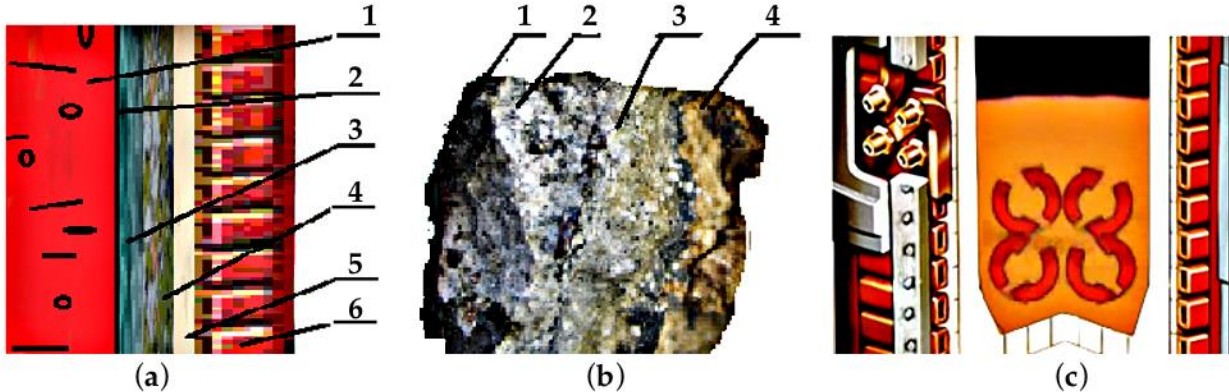

**Figure 3.** Induction furnace lining layers: (**a**) layer layout: 1—liquid melt, 2—slagged layer, 3—sintered layer, 4—intermediate layer, 5—unaltered (loose) layer, 6—furnace inductor; (**b**) knocked-out lining sample: 1—slagged layer, 2—sintered layer, 3—intermediate layer, 4—virtually unchanged layer consisting of quartzite bonded with boric acid; (**c**) scheme of melt mixing.

B. P. Platonov, in the study of the metallurgical processes occurring during the smelting of synthetic iron in an induction melting furnace of industrial frequency, also described the presence of three zones along its thickness (Figure 3) [27]. The first zone is little changed (buffer), adjacent to the inductor; the second is transitional, in which the lining mass is not completely sintered; the third, adjoining the melt, is completely sintered. The phase composition of the zones is different, so the lining, which withstands 300 heat cycles in the middle of the furnace, has the following composition: the first zone consists of 91.5% tridymite (the rest is a vitrified mass); the second (intermediate) contains 73% quartzite, 12% cristobalite and the rest tridymite; the third zone is 100% quartzite with boric acid. Figure 3a shows the layout of the layers of the lining, Figure 3b shows a sample of the knocked-out lining, and Figure 3c shows the melt mixing scheme.

## 2. Materials and Methods

### 2.1. Formulation of the Problem

Thus, the intensity of the transition to tridymite and cristobalite depends not only on the technological parameters of the production process but also on the properties of the quartzites used [28]. Until recently, the initial scheme characterizing the phase transformations of quartzite during heating was the Fenner diagram [29], according to which, when quartzite is heated, the tridymite phase is first formed followed by cristobalite. However, many studies have emerged that refute this pattern. This is due to the imperfection of the methods of physicochemical analysis used by Fenner. The emergence of the method of X-ray diffraction analysis and the development of physical and chemical methods, in the 1960s to 1970s, made it possible to clarify the phase transformations of quartzite. Since 1961, when conducting studies of phase transformations of quartzite during heating, tridymite has not been found [30–32]. V. P. Pryanishnikov proposed his version of the modified diagram of the $SiO_2$ system [33].

Other researchers have identified the pattern of the appearance of cristobalite first followed by tridymite [34–40]. The transformation of quartz into cristobalite begins at a temperature of 1000 °C, and with long exposure, it intensifies at a temperature in the

range of 1250–1450 °C [41–44]. This is also affected by the impurities contained in the original quartzite [45]. Microscopic studies have shown that the transformation of quartz into cristobalite occurs from the grain surface; for large grains (2–3 mm), it also occurs along cracks formed during heat treatment, and the new cristobalite phase formed exactly resembles the shape of a quartz grain. In addition, the condition for the presence of a mineralizer to obtain tridymite was established [41,46].

This process is also influenced by impurities in the quartzite itself [47–49]. Recently, research works have been presented on the preliminary processing of the original quartzite, which makes it possible to obtain the necessary crystalline phase. These are technologies for cleaning quartzite using magnetic separation, flotation, microwave treatment, thermal shock and chlorination, changing the grain composition [50]. The simplest way to obtain first tridymite and only then cristobalite is the use of impurities or mineralizers, but this leads to a decrease in the refractoriness of quartzite. A particularly urgent task is to obtain the necessary phase of quartzite, which is used to produce the lining of induction crucible melting furnaces, since intensive mixing of the melt occurs in them (the mixing scheme is shown in Figure 3c). The most favorable phase composition of quartzite in a lining based on it was considered to be tridymite, which retains a constant volume at 840–1470 °C for a long time and thereby ensures the high durability of the lining and hence the efficiency of the melting furnace [51]. However, since the 2000s, scrap iron used in metal filling to smelt synthetic pig iron has virtually disappeared. For this reason, foundries are forced to smelt synthetic iron from a metal charge consisting of 98% of one steel scrap, which requires melting at a temperature of 1470 to 1550 °C, instead of 1450 °C, with a short-term increase to 1570 °C. The durability of the lining drops sharply, from 350 to 250 melts. This leads to an increase in furnace downtime associated with its relining. Thus, an induction crucible melting furnace with a three-shift operation and a lining life of 350 melts (productivity of 3 melts per shift) requires 2–2.5 relinings per year, or 5 days. With a durability of 220 melts, 7 days are needed, which means that 18 melts or 18 tons of cast iron are lost, not to mention the forced downtime for smelters.

In addition, since 2000, cases of rapid wear of the lining due to the fault of the quartzite pretreatment technology have become more frequent. Pretreatment with quartzite should be designed in order to obtain the required moisture content of not more than 0.3% (many enterprises buy quartzite with a moisture content of 3%), using the following temperature regime: heating to the range of 800–900 °C and subsequent exposure for 8–10 h using stainless steel containers. This technology has been used since the beginning of the use of industrial frequency induction crucible melting furnaces. It guarantees, after the sintering of the lining, the formation of the tridymite phase, but it cannot provide the necessary refractoriness since the smelting of synthetic iron is carried out at elevated temperatures from 1500 to 1550 °C. For this reason, research work is constantly underway to increase the resistance of the acid lining [52].

This work describes research into the effect of quartzite pretreatment on the formation of its phases. Pretreatment consisted of drying samples of the original quartzite at temperatures of 200 and 800 °C for 1 h (to remove moisture) and further cooling each sample to room temperature. Then, these quartzite samples were exposed to temperatures of 200, 400, 600, 870, 1000, 1200, 1470 and 1550 °C. As a result, the conditions for the preliminary processing of the original Pervouralsky quartzite are established, during which, during its further use, a phase of tridymite or cristobalite can be obtained. In the future, these studies will make it possible to develop a technology for manufacturing a high-temperature lining for an induction crucible melting furnace of industrial frequency, which ensures its high durability.

### 2.2. Description of the Field Experiment

The study was carried out for Pervouralsky quartzite grade PKMVI-1, with a moisture content of 3.5%, supplied by OJSC "DINUR" according to technical specifications (TS) 1511-022-00190495-2003 (Figure 4b). According to the supplier, the finished product mainly

contained quartz, but it also contained impurities of chalcedony, carbonates, opal and clay minerals, with a low concentration of iron oxides and high dispersion. However, according to the current TS, the supplied quartzite had the chemical composition presented in Table 2 (the content of other impurities was not given). In addition, the following grain composition was guaranteed, in mass fractions:

- Remaining on grid No. 2, including—8–14;
- Remaining on the grid No. 3.2, no more—5;
- Passage through grid No. 05, including—46–51;
- Passage through grid No. 01—27–32.

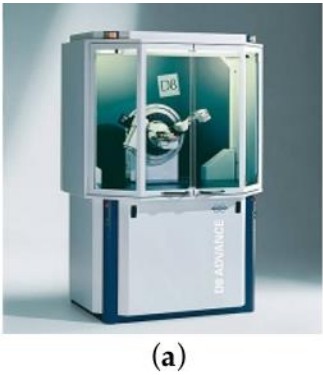 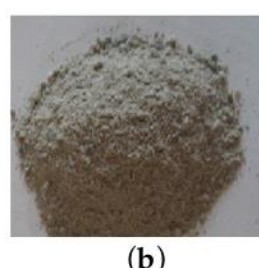 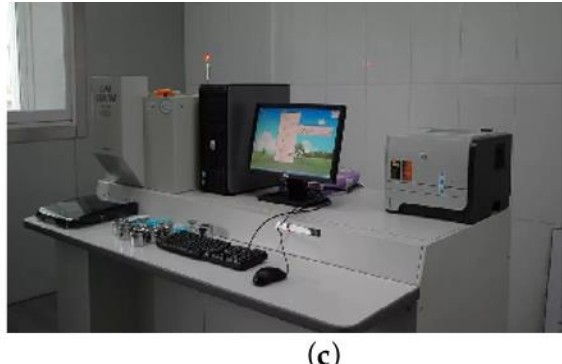

**Figure 4.** Examination equipment and material: (**a**) diffractometer Bruker D8 ADVANCE; (**b**) PKMVI-1 quartzite; (**c**) Shimadzu XRF-1800 X-ray wave-dispersive spectrometer.

**Table 2.** The chemical compositions of quartzite grade PKMVI-1.

| Chemical Composition of PKMVI-1 Quartzite | Content, % | | | | | | | | | | |
|---|---|---|---|---|---|---|---|---|---|---|---|
| | $SiO_2$ | $Al_2O_3$ | CaO | MgO | $TiO_2$ | $Fe_2O_3$ | $P_2O_5$ | MnO | CaO | $Na_2O$ | $K_2O$ |
| According to the TS | 97.5 | 1.1 | - | - | - | 0.6 | - | - | - | - | - |
| Spectrometer dried at 200 °C | 96.74 | 0.862 | 0.097 | 0.027 | 0.275 | 1.02 | 0.015 | 0.038 | 0.098 | 0.12 | 0.584 |

To study the phase composition, the X-ray diffraction method was used, adopting a BRUKER D8 ADVANCE diffractometer (Bruker Corporation, Billerica, MA, USA) equipped with Bragg–Brentano focusing and an HTK 16 high-temperature camera. An X-ray tube with a copper anode was used; the diffraction spectrum was recorded with a high-speed position-sensitive detector, VÅNTEC-1. The survey was carried out at scanning angles 2Θ = 10–90° with a step of 0.007, and the duration of the survey was 1 h (Figure 4a).

Chemical analysis was carried out using the method of fundamental parameters using a Shimadzu XRF-1800 X-ray wave-dispersive spectrometer (Shimadzu, Kyoto, Japan) (Figure 4c). The device was equipped with collimators and a built-in digital camera, and the sample rotation speed was 60 rpm.

The research methodology included the following steps:

1. We heated a portion of raw quartzite to a temperature of 800 °C, allowed exposure for 1 h and cooling to room temperature, and then subjected it to subsequent heating with exposure at each point to remove the card and determine the parameters of the crystal lattice, at 200, 400, 600, 870, 1000, 1200, 1470, 1550 °C;
2. We heated the next portion of raw quartzite to a temperature of 200 °C, allowed exposure for 1 h and cooling to room temperature, and then determined its chemical composition and subjected it to subsequent heating with exposure to remove the card and determine the parameters of the crystal lattice, at 200, 400, 600, 870, 1000, 1200, 1470, 1550 °C.

The choice of study temperatures was based on the following:

- 200 °C—approximate value of the free moisture removal temperature;
- 400 °C—the appearance of cells in the structure of quartzite with a lower molecular weight (M = 59.2 g/mol), established in [53];
- 600 °C—an intense phase transformation occurs with the release of heat, which was established in [54];
- 1025 °C—furnace drain temperature;
- 1470–1550 °C—melting mode;
- 1550–1570 °C—carrying out the operation of alloying and modification.

## 3. Results

Chemical analysis, carried out by the method of fundamental parameters using an X-ray fluorescence wave-dispersive spectrometer, the Shimadzu XRF-1800 (Shimadzu, Kyoto, Japan), made it possible to establish the actual chemical composition of PKMVI-1 quartzite dried at 200 and 800 °C (Table 2). This showed that there were impurities in the quartzite, in the amount of 1.254%, which could affect the process of phase formation during its heating and application.

At the first stage, a study was carried out on quartzite subjected to calcination at 800 °C and cooled to 30 °C. The diffraction pattern shown in Figure 5 indicates that the lattice structure consisted of two elementary cells, characterized by the cards 01-083-2187 and 01-070-7344.

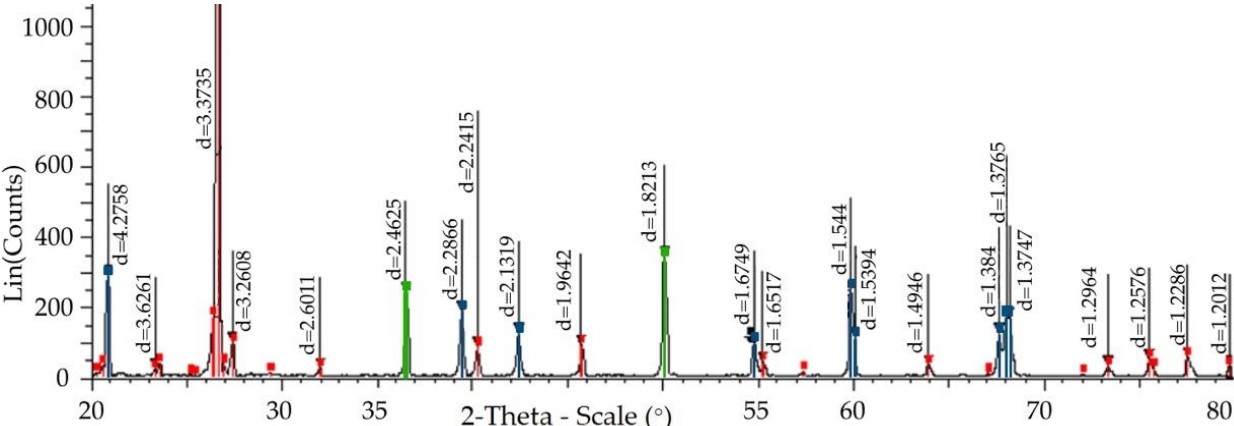

**Figure 5.** The X-ray diffraction patterns of quartzite taken at 30 °C, where green indicates peaks for 01-083-2187, and peaks for 01-070-7344 are marked in blue.

The interpretation of the changes in the structure of quartzite was carried out using the instrument's program, which had a bank of necessary data in the form of cards of elementary cells with their characteristics. Table 3 shows the characteristics of the cards used during the first and second stages of the study.

Based on the obtained data for interplanar distance, the percentage of elementary cells was determined; the interplanar distance d ≥ 5% was taken into account (the program made it possible to determine which elementary cells corresponded to the interplanar distances). The data obtained during the first and second stages of the study (separated by the symbol "/") are shown in Table 4.

At a temperature of 200 °C, there was an increase in the proportion of elementary cells of quartzite with card 01-083-2197 to 86% and a decrease in the proportion of elementary cells with card 01-070-7344 to 14%.

At a temperature of 400 °C, the structure of quartzite consisted only of elementary cells with card 01-083-2197.

**Table 3.** The characteristics of the cards used in the research process.

| Indicators | Patent: 00-012-0708 SiO$_2$ Silicon Oxid Quartz Hexagonal P3221 (154) | Patent: 01-071-0911 SiO$_2$ Silicon Oxid Quartz Hexagonal P6222 (180) | Patent: 00-005-0490 SiO$_2$ Silicon Oxid Quartz Hexagonal P3121 (152) | Patent: 01-083-2187 SiO$_2$ Silicon Oxid Quartz Hexagonal P3221 (152) | Patent: 01-070-7344 SiO$_2$ Silicon Oxid Quartz Hexagonal P3221 (154) | Patent: 02-002-0278 SiO$_2$ Silicon Oxid Cristobalite Cubic Fd-3m (227) | Patent: 01-011-0695 SiO$_2$ Silicon Oxid Tetragonal P41212 (92) | Patent: 01-085-0621 SiO$_2$ Silicon Oxid Cristobalite Cubic P213 (198) | Patent: 01-071-0032 SiO$_2$ Silicon Oxid Tridymite Monoclinic Cc (9) | Patent: 00-018-1170 SiO$_2$ Silicon Oxid Tridymite Monoclinic Cc (9) |
|---|---|---|---|---|---|---|---|---|---|---|
| a, (Å) | 4.994 | 5 | 4.913 | 4.965 | 4.915 | 7.12 | 4.971 | 7.16 | 18.494 | 18.504 |
| b, (Å) | 5.438 | 5.49 | 5.405 | 5.424 | 5.406 | - | - | - | 4.991 | 5.006 |
| Mol. weight, (g/mol) | 60.08 | 52.87 | 60.08 | 60.08 | 60.08 | 60.08 | 60.08 | 60.08 | 60.08 | 60.08 |
| Volume, (Å$^3$) | 117.45 | 118.86 | 112.98 | 115.79 | 113.09 | 360.94 | 170.95 | 367.06 | 2110.15 | 2215.08 |
| D$_x$, (g/cm$^3$) | 2.548 | 2.216 | 2.649 | 2.58 | 2.647 | 2.211 | 2.335 | 2.174 | 2.270 | 2.254 |
| c, (Å) | - | - | - | - | - | - | - | - | 26.832 | 23.845 |

**Table 4.** The percentages of elementary cells characterized by the corresponding cards.

| Temperature, (°C) | The Content of Elementary Cells, %, 1/2 Stage of the Study | | | | | | | | | |
|---|---|---|---|---|---|---|---|---|---|---|
| | 00-012-0708 | 01-071-0911 | 01-083-2187 | 01-070-7344 | 01-071-0032 | 00-018-1170 | 01-085-0621 | 00-005-0490 | 01-011-0695 | 02-002-0278 |
| 25 (initial) | 58 | | | 42 | | | | | | |
| 800/200 | 10/0 | 90/0 | 0/67 | 0/23 | 0/10 | | | | | |
| 30/25 | 0/6 | | 65/69 | 35/10 | | | | 0/15 | | |
| 200 | 0/4 | | 86/74 | 14/15 | | | | 0/7 | | |
| 400 | | 0/8 | 100/92 | | | | | | | |
| 600 | 61/0 | 0/68 | 20/32 | | 19/0 | | | | | |
| 870 | | 24/0 | | | 0/87 | 76/0 | | | 0/13 | |
| 1000 | | 20/0 | | | 0/89 | 80/0 | | | 0/11 | |
| 1200 | | 19/0 | | | 0/89 | 81/0 | | | 0/11 | |
| 1470 | | 18/0 | | | 0/89 | 82/0 | | | 0/11 | |
| 1550 | | 18/0 | | | 0/79 | 73/0 | 9/0 | | 0/9 | 0/12 |

Heating to 600 °C led to the appearance of a unit cell of tridymite card 01-071-0032 in the amount of 19%.

At a temperature of 870 °C, new elementary cells of tridymite 00-018-1170 appeared in the amount of 76% and 24% of cells with 01-071-0911.

Further heating to 1000, 1200 and 1470 °C only showed an increase in the proportion of tridymite, and at 1400 °C, the content of these cells was 82%.

At a temperature of 1550 °C, elementary cells of cristobalite (card 01-085-0621) appeared in the structure of quartzite in the amount of 9%, the number of elementary cells of tridymite 00-018-1170 was 73%, and cells with 01-071-0911-18 were present.

The diffraction pattern obtained at 600 °C is shown in Figure 6a, while the diffraction pattern of quartzite at a temperature of 1550 °C is shown in Figure 6b.

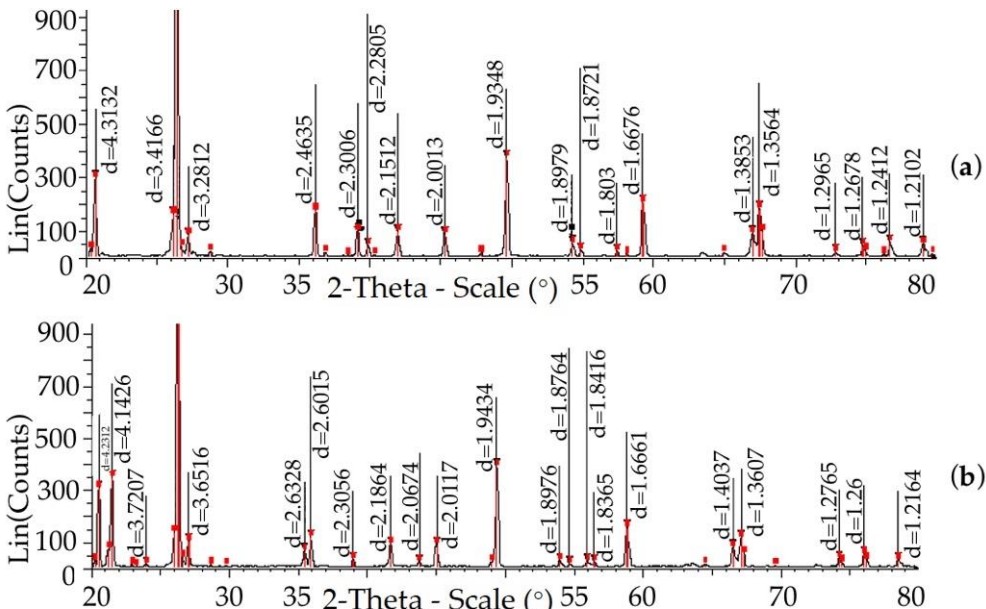

**Figure 6.** The X-ray diffraction patterns of quartzite: (**a**) obtained at 600 °C; (**b**) obtained at 1550 °C.

At the second stage, a study was carried out on quartzite subjected to drying at 200 °C and cooling to 25 °C.

The diffraction pattern shown in Figure 7 gives information about the parameters of the structure of the quartzite dried at 200 °C and cooled to 25 °C. The lattice consisted of four varieties of elementary cells of quartzite (cards 00-012-0708, 01-070-7344, 00-005-0490 and 01-083-2187).

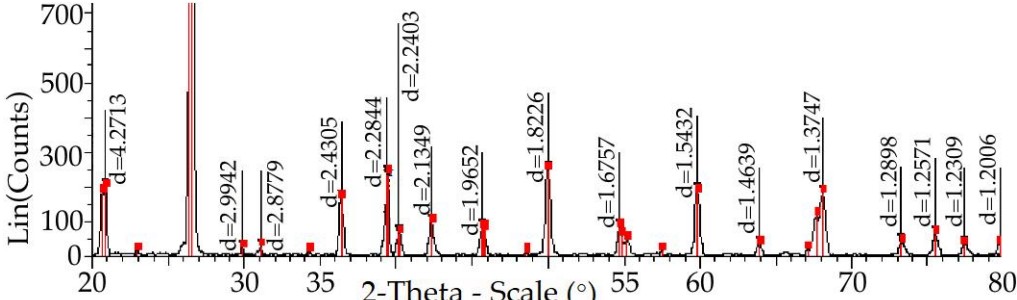

**Figure 7.** The diffractogram of the structure of quartzite dried at 200 °C and cooled to 25 °C.

At a temperature of 200 °C, there was only a change in the content of these unit cells.

At a temperature of 400 °C, the lattice parameters underwent changes due to the fact that the elementary cells with the cards 00-012-0708, 00-005-0490 and 01-070-7344 disappeared, and the structure of the quartzite consisted of two varieties of elementary cells with cards 01-083-2187 and 01-071-0911.

At 600 °C, there was only a change in the content of these cells in the structure.

Heating to 870 °C led to an increase in the content of the unit cells with card 01-071-0911 to 87% and the appearance of a unit cell of cristobalite, with card 00-011-0695, in the amount of 13%.

At temperatures of 1000, 1200 and 1470 °C, there were no significant changes in the structure of quartzite, and its parameters practically did not change. Heating to 1550 °C led to the appearance of another unit cell of cristobalite with 00-002-0278 in the amount of 13%. The diffraction pattern of the structure of quartzite at 870 °C is shown in Figure 8a, and that at a temperature of 1550 °C is shown in Figure 8b.

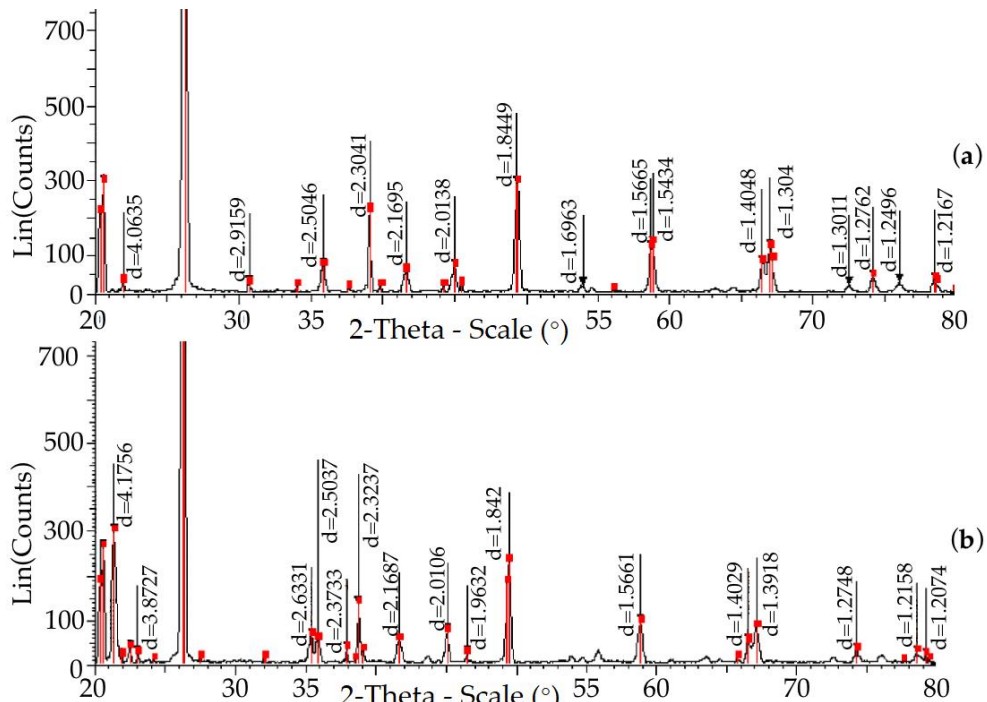

**Figure 8.** The X-ray diffraction patterns of quartzite: (**a**) taken at 870 °C; (**b**) removed at 1550 °C.

The averages of the lattice parameters—interplanar distance ($d_{avg}$), volume of the unit cell ($V_{avg}$), density ($D_{avg}$) and molecular weight ($M_{avg}$)—obtained from studies of quartzite calcined at a temperature of 800 °C, cooled to ambient temperature and subsequently heated from 30 to 1550 °C, were determined on the basis of the obtained interplanar distances for each diffraction pattern, and they are given in Table 5.

**Table 5.** The results of the study of quartzite calcined at a temperature of 800 °C, cooled to ambient temperature and subsequently heated from 30 to 1550 °C.

| Parameter Gratings | Temperature, °C | | | | | | | | | | |
|---|---|---|---|---|---|---|---|---|---|---|---|
| | 25 Raw | 800 | 30 Dry | 200 | 400 | 600 | 870 | 1000 | 1200 | 1470 | 1550 |
| $d_{avg}$, Å | 2.676 | 3.012 | 2.757 | 2.901 | 2.946 | 3.008 | 3.055 | 3.105 | 3.126 | 3.156 | 3.216 |
| (change, %) | - | **12.56** | **3.04** | **8.42** | **10.09** | **12.39** | **14.14** | **16.02** | **16.83** | **17.93** | **20.16** |
| $V_{avg}$, Å$^3$ | 115.4 | 129.85 | 114.83 | 115.41 | 115.79 | 647.47 | 1653.02 | 1722.83 | 1742.69 | 1767.17 | 1606.96 |
| (change, %) | - | **12.52** | **−0.49** | **0.0** | **0.34** | **561.06** | **1432.42** | **1492.92** | **1510.13** | **1531.34** | **1392.51** |
| $D_{avg}$, g/cm$^3$ | 2.59 | 2.249 | 2.601 | 2.592 | 2.585 | 2.502 | 2.269 | 2.266 | 2.265 | 2.265 | 2.258 |
| (change, %) | - | **−13.17** | **+0.42** | **+0.08** | **−0.19** | **−3.4** | **−12.41** | **−12.51** | **−12.54** | **−12.54** | **−12.81** |
| $M_{avg}$, g/mol | 60.08 | 51.01 | 60.08 | 60.08 | 60.08 | 60.08 | 58.38 | 58.63 | 58.7 | 58.79 | 58.79 |
| (change, %) | - | - | - | - | - | - | **−2.83** | **−2.41** | **−2.3** | **−2.15** | **−2.15** |

The first significant changes occurred at a temperature of 600 °C and were characterized by an increase in the average interplanar distance in comparison with the dried sample (column in Table 5, 30 dry) by 9%, the average volume increased by 5.63% and there was a slight decrease in density values of 4%. This occurred due to the appearance of a tridymite elementary cell, card 01-071-0032, in the amount of 19%.

At 870 °C, a new unit cell of tridymite 00-018-1170 appeared, in the amount of 76%. As a result, there was an increase in the average interplanar distance compared with the dried sample (column in Table 5, 30 dry) by 10.8%, the average volume by 14.39% and a decrease in density by 12.8% and molecular weight by 2.2%. With further heating, only

an increase in the proportion of the content of elementary cells of tridymite occurred, card 01-071-0032 up to 80%.

At a temperature of 1470 °C, the following changes occurred in the lattice structure: an increase in the average interplanar distance compared with the dried sample (column in Table 5, 30 dry) by 14.4% and an increase in the average volume by 15.39% without changes in density and molecular weight.

At 1550 °C, elementary cells of cristobalite appeared in the lattice structure (card 01-085-0621) in the amount of 9%. This led to the following observations: the average interplanar distance compared from the dried sample (column in Table 5, 30 dry) increased by 16.6%, and the average volume increased by 1400% without changes in density and molecular weight.

The results of the study of transformations in the structure of quartzite subjected to drying at 200 °C followed by cooling to ambient temperature and subsequent heating from 25 to 1550 °C are presented in Table 6.

**Table 6.** Changes in the lattice parameters of quartzite during heating from 25 to 1550 °C with pre-drying at 200 °C.

| Parameter Gratings | Temperature, (°C) | | | | | | | | | | |
|---|---|---|---|---|---|---|---|---|---|---|---|
| | 25 Raw | 200 | 25 Dry | 200 | 400 | 600 | 870 | 1000 | 1200 | 1470 | 1550 |
| $d_{avg}$, Å | 2.676 | 2.908 | 2.814 | 2.834 | 2.892 | 2.791 | 2.928 | 2.98 | 2.992 | 3.038 | 3.262 |
| (change, %) | - | 8.67 | 5.16 | 5.9 | 8.08 | 4.31 | 9.41 | 11.34 | 11.79 | 13.54 | 21.99 |
| $V_{avg}$, Å$^3$ | 115.4 | 116.35 | 119.1 | 116.55 | 115.97 | 117.47 | 125.86 | 124.06 | 124.19 | 124.04 | 143.65 |
| (change, %) | - | 0.82 | 3.2 | 0.99 | 0.49 | 1.79 | 9.06 | 7.5 | 7.62 | 7.49 | 24.48 |
| $D_{avg}$, g/cm$^3$ | 2.590 | 2.552 | 2.597 | 2.552 | 2.558 | 2.333 | 2.292 | 2.291 | 2.229 | 2.229 | 2.227 |
| (change, %) | - | −1.5 | +0.27 | −1.5 | −1.24 | −9.93 | −11.51 | −11.54 | −13.93 | −13.93 | −14.01 |
| $M_{avg}$, g/mol | 60.08 | 60.08 | 60.08 | 60.08 | 59.54 | 55.16 | 53.91 | 53.66 | 53.67 | 53.66 | 54.41 |
| (change, %) | - | - | - | - | −0.9 | −8.19 | −10.27 | −10.59 | −10.67 | −10.69 | −9.44 |

The first significant changes occurred at a temperature of 400 °C and were characterized by an increase in the average interplanar distance compared with the dried sample by 2.8%, a decrease in the average volume by 2.5% and insignificant decreases in the density and molecular weight. The reason for these results was the appearance of the quartzite phase, characterized by the card 01-071-0911-8.

At a temperature of 600 °C, the proportion of these unit cells increased to 68%. This led to the fact that the average lattice volume decreased by 1.4% compared with the dry sample (column in Table 6, 25 dry) and the average density and molecular weight decreased by approximately 10.2%.

At a temperature of 870 °C, elementary cells of cristobalite appeared, characterized by card 00-011-0695, in an amount of 13%. As a result, drastic changes occurred in the structure of the lattice: the average interplanar distance compared with the dried sample (column in Table 6, 25 dry) increased by 4%, the average volume increased by 5.7% and the average density decreased by 10.2%. With a further increase in temperature, changes occurred due to a decrease in the content of these cristobalite cells to 11%, and at 1470 °C, the average interplanar distance of the lattice compared with the dried one (column in Table 6, 25 dry) increased by 8%; the average volume increased by 6.4%, and the average density decreased by 4.2%.

At a temperature of 1550 °C, another unit cell of cristobalite appeared in the amount of 12%, with card 00-002-0278. This resulted in a dramatic change: the average interplanar spacing increased by 15.9% and the average volume by 12%.

Figure 9 shows diagrams of the phase changes in quartzite with different technologies designed to reduce its moisture content to ≤0.3%. Phase transformations occurring

in quartzite when it was heated from 30 to 600 °C (Figure 9a)—subjected at the same time to preliminary treatment that consisted of calcination at a temperature of 800 °C—proceeded in accordance with the Fenner diagram. The structure consisted of elementary cells corresponding to two phase states: (1) Quartz low and (2) Quartz high.

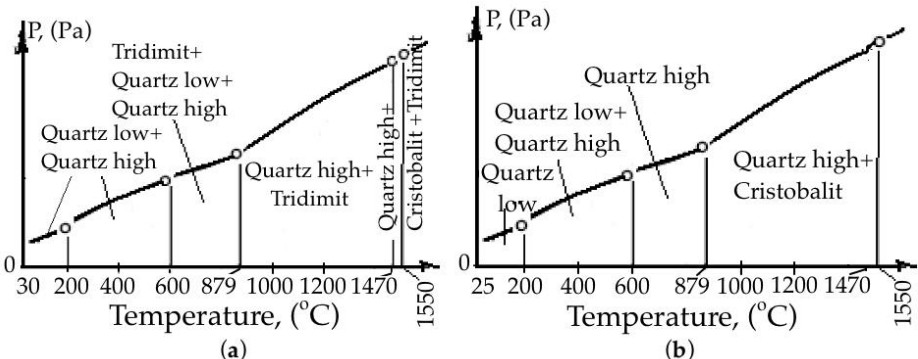

**Figure 9.** Phase transformations in quartzite during its heating: (**a**) heat treatment at 800 °C; (**b**) heat treatment at 200 °C.

When heated from 600 to 879 °C, despite the absence of a mineralizer, along with Quartz low and Quartz high, the Tridimit phase appeared, and above 879 up to 1470 °C, only Quartz high and Tridimit remained. This phase state maintained a constant volume at temperatures of 840–1470 °C for a long time and provided (this has been shown by the practice of operating industrial induction melting furnaces) high lining durability in the amount of 300–350 melts, but only when using melting temperature regimes of no more than 1500 °C. In the temperature range from 1470 to 1550 °C, the average interplanar distance increased by 2%.

Above 1470 °C, the cristobalite phase began to appear, and the quartzite structure consisted of three phases: Quartz high, Tridimi and cristobalite. At temperatures above 1470 °C, the material had a lower volumetric expansion, which led to an increase in lining cracking and a decrease in its durability.

Thus, the use of this preliminary temperature treatment of quartzite when using melting temperatures above 1500 °C leads to the formation of a phase state, which leads to an increase in downtime due to an increase in the number of relinings and reduces the efficiency of the production of foundry products.

Phase transformations that occur in quartzite when it is heated from 30 to 600 °C (Figure 9b)—subjected at the same time to preliminary treatment, which consists of drying at a temperature of 200 °C—proceed in accordance with the Fenner diagram. The structure consists of elementary cells corresponding to the phase states Quartz low and Quartz high.

When heated from 600 to 879 °C, the structure consists only of the Quartz high phase, and above 879 °C, it forms the cristobalite phase, the melting point of which is 1713 °C, while, in Tridimi, it is 1670 °C, which increases the refractoriness. In this phase, the state is characterized by volume constancy at temperatures of 840–1470 °C, and in the temperature range of 1470–1550 °C, there is an increase in the average interplanar distance by 7%. For this reason, quartzite, which has a phase composition of Quartz high and cristobalite, is more resistant to the thermal expansion of the lining during melting, draining and loading (840–1550 °C), providing (this has been shown by the practice of operating induction melting industrial furnaces), over a long period of time, high resistance in the amount of 300–350 melts.

Thus, the obtained schemes of quartzite phase changes (Figure 9) make it an appropriate choice for pretreatment, with the subsequent manufacture of the lining of various melting units (electric arc, induction, reverberatory and rotary melting furnaces).

## 4. Conclusions

The results of this study allowed us to establish the conditions for the preliminary preparation of quartzite, which make it possible to obtain the Tridimit and cristobalite phases in the material under the influence of temperatures during the sintering of the lining based on it and further modes of smelting synthetic iron.

Despite the presence of an impurity in quartzite in the amount of 1.254%, its preliminary calcination at a temperature of 800 °C and further cooling makes it possible to simultaneously obtain three phases on its basis in the process of sintering the crucible lining of an induction furnace: Quartz high + Tridimit + cristobalite. Tridymite begins to appear at 600 °C; hence, no mineralizer is required. Cristobalite appears only at a temperature of 1550 °C.

Moreover, the results of the study showed that tridymite retains its properties up to a temperature of 1470 °C, and an acid lining having such a phase state is highly resistant, but under the condition of smelting, synthetic irons no higher than 1450 °C are required; this requirement is specified in the operating instructions for the induction furnace. The smelting of pig iron on scrap steel requires the use of temperature regimes of 1470–1570 °C, for which the use of a different phase state of quartzite is necessary, cristobalite, which can withstand temperatures up to 1620 °C.

Tridymite, in the temperature range of 1200–1550 °C (these are the operating temperatures during smelting), allows for volumetric changes in the range of 9% and cristobalite in the range of 15%. This means that the lining based on cristobalite is more resistant to extreme temperature changes.

Preliminary treatment of quartzite PKMVI-1, which consists of its drying at 200 °C, makes it possible to obtain the Quartz high phase and the cristobalite phase in the material during the further preparation of the lining mass and the packing and sintering of the lining of the melting crucible furnace of industrial frequency. Cristobalite appears at a temperature of 870 °C.

Thus, the preliminary preparation of quartzite, which consists of drying it at 200 °C, will make it possible to obtain the Quartz high phase and the cristobalite phase in the material, achieving high durability of the lining during the smelting of synthetic iron from one steel scrap, and improving the efficiency of induction crucible furnaces of industrial frequency and the production of foundry products.

One of the future directions resulting from the study described is the study of the influence of the moisture state on the properties of quartzite when it is heated.

**Author Contributions:** Conceptualization, V.A.K., A.I.C. and V.S.T.; Data curation, V.A.K., A.I.C., V.V.K., V.S.T., S.O.K. and R.B.S.; Formal analysis, V.A.K., A.I.C., V.V.K., V.S.T., S.O.K. and V.V.T.; Investigation, V.A.K., A.I.C., V.V.K., V.S.T., S.O.K., K.A.B. and R.B.S.; Methodology, V.A.K., V.V.K., K.A.B. and V.V.T.; Project administration, V.A.K., A.I.C. and V.S.T.; Resources, V.A.K. and S.O.K.; Supervision, V.A.K., A.I.C. and V.S.T.; Validation, A.I.C., V.V.K., V.S.T., S.O.K., K.A.B., R.B.S. and V.V.T.; Visualization, V.V.K., R.B.S. and V.V.T.; Writing—original draft, V.A.K., A.I.C., V.V.K., V.S.T., K.A.B., R.B.S. and V.V.T.; Writing—review & editing, V.A.K., A.I.C., V.V.K., V.S.T., S.O.K., K.A.B., R.B.S. and V.V.T. All authors have read and agreed to the published version of the manuscript.

**Funding:** This research received no external funding.

**Institutional Review Board Statement:** Not applicable.

**Informed Consent Statement:** Not applicable.

**Data Availability Statement:** Not applicable.

**Conflicts of Interest:** The authors declare no conflict of interest.

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
