# Peer review of "Increasing the Efficiency of Foundry Production by Changing the Technology of Pretreatment with Quartzite"

_metals, doi:10.3390/met12081266_

Round 1

Reviewer 1 Report

This work reports increase of the efficiency of the production of foundry products by changing the technology of pretreatment of quartzite, which is indeed novel. However, this manuscript only presents some experimental results, and there is no deep excavation of the principles. Therefore, the reviewer think that it is not published in its current form. I suggest the authors to clarify and discuss more about the essence behind the experimental phenomenon, and then resubmit the revised manuscript.

Author Response

Dear Reviewer 1,

Thank you for your remarks on our paper, so we’ve made changes in the article.

Remark 1: This work reports increase of the efficiency of the production of foundry products by changing the technology of pretreatment of quartzite, which is indeed novel. However, this manuscript only presents some experimental results, and there is no deep excavation of the principles. Therefore, the reviewer think that it is not published in its current form. I suggest the authors to clarify and discuss more about the essence behind the experimental phenomenon, and then resubmit the revised manuscript.

Reply: Thank you for the comment. Studies have been carried out on two options for pre-treatment of quartzite in order to remove moisture from it and their effect on the formation of tridimite and cristobalite phases.

To increase the efficiency of grey iron castings using industrial frequency induction crucible furnaces, the largest amount of steel scrap needs to be used in scrap metal filling, but this entails an increase in melting operating temperatures, and therefore reduces the stability of the lining and increases its downtime. For this reason, the preliminary preparation conditions of the raw quartzite have been determined, allowing after the production of the lining on its basis, the packing and sintering of the lining to obtain in it a phase of cristobalite that withstands higher temperatures, higher wear and tear of the crucible’s slag belt and increasing metal movement intensity.

With best regards,

Dr. Vadim Tynchenko

Reviewer 2 Report

The paper addresses the problem of the efficiency of foundry production. It proposes a technology for the pre-treatment of original quartzite, which is supposed to ensure the formation of a phase state that successfully withstands elevated temperatures for a long time. The research has been done experimentally and the authors could establish the promising pre-treatment conditions that offer high durability.    The paper is generally well written. It would be of interest to the research community in this field of work. However, certain minor aspects should be addressed before the paper can be suggested for publishing:  1) Generally speaking, the paper is well structured. However, the introduction is written on nearly 5 pages out of 12.5 pages of the overall length. This is simply too much and the authors should reconsider either to shorten the introduction, or divide it into two suitable sections - maybe an introduction that explains the motivation for the work, and another section that represents the previous work in the field (state-of-the-art).  2) The quality of some figures (e.g. 2, 3 and the right-hand side of 4) is under par and should be improved.  3) The conclusions should also mention some possible directions of future research. 

Author Response

Dear Reviewer 2,

Thank you for your remarks on our paper, so we’ve made changes in the article.

Remark 1: The paper addresses the problem of the efficiency of foundry production. It proposes a technology for the pre-treatment of original quartzite, which is supposed to ensure the formation of a phase state that successfully withstands elevated temperatures for a long time. The research has been done experimentally and the authors could establish the promising pre-treatment conditions that offer high durability. The paper is generally well written. It would be of interest to the research community in this field of work. However, certain minor aspects should be addressed before the paper can be suggested for publishing: 1) Generally speaking, the paper is well structured. However, the introduction is written on nearly 5 pages out of 12.5 pages of the overall length. This is simply too much and the authors should reconsider either to shorten the introduction, or divide it into two suitable sections - maybe an introduction that explains the motivation for the work, and another section that represents the previous work in the field (state-of-the-art).  2) The quality of some figures (e.g. 2, 3 and the right-hand side of 4) is under par and should be improved.  3) The conclusions should also mention some possible directions of future research.

Reply: Thank you for the comment. We have reduced the introduction (line 156).

The drawings are made with photos of equipment in the current technological process, so the clarity of contrast and brightness are not always optimal. But we have tried to corrected the quality of figures 2,3,4.

We have added possible directions of future research in conclusion section (line 470).

With best regards,

Dr. Vadim Tynchenko

Reviewer 3 Report

Currently, the world is striving to implement the idea of the so-called Zero Waste, which is a response to environmental damage caused by the consumer's lifestyle. The increasing amount of waste generated and the consumption of natural resources too quickly is driving many people to "switch to Zero Waste". This idea does not bypass the foundry industry, where apart from recycling, for example, metal scrap, we are also dealing with the saving of natural resources (refreshing and reclamation process of moulding sands) and the production of ecological binders. Cast metal manufacturers rely on different types of furnace technologies. Although blast furnaces are often associated with metal working, they’re actually used for extracting iron and other metals from ores. Manufacturing foundries need other furnaces to take metal alloys and additives and make them into certain grades of cast metal. The most common four furnaces in use at a manufacturing foundry: induction furnaces; crucible furnaces; cupola furnaces; electric arc furnaces. The purpose of the lining is to prevent the furnace from overheating and wasting energy. This can be achieved by using dense bricks and mortars, insulating bricks, refractory concretes or ceramic fiber materials, usually in the form of modules. Materials with a low thermal capacity have the advantage that they heat up and cool down faster, while materials with a higher thermal mass are more durable and wear-resistant.

Companies offer an extremely wide range of materials for lining blast furnaces depending on the specific installation conditions and loads. Fired at temperatures up to 1700°C, corundum bricks are characterized by high resistance to abrasion and changes in thermal conditions, thanks to which they withstand mechanical stress in the most loaded, upper part of the furnace. In the reduction zones, special chamotte and andalusite fittings are used, which ensure exceptional resistance to the reducing environment. The high temperature and pressure generated by the charge put an enormous strain on the melting zone of the furnace. Ring elements made of chemically / ceramically bonded andalusites have been developed for highly stressed areas. The bottom of the furnace, the hearth, is covered with many layers of carbon brick. Thanks to the use of low-iron and low-iron chamotte grades and good heat resistance, which is combined with high-quality sintered mullite (with exceptional corrosion resistance), the "ceramic cup" of the furnace can be protected from the effects of pig iron. There is an increasing demand for innovative heat-resistant systems that increase the efficiency of entire plants. In addition to the high stresses generated by the charge, atmosphere and temperature variations, the lining is also negatively affected by the increased flow of materials, thereby causing an additional increase in the friction level. In modern foundries, the channels for ingot molds are lined with the highest quality refractory products. The produced steel must be free from non-metallic inclusions and have a stable composition and homogeneous structure. In the case of high-quality high-alloy steels, such as manganese steels, a material with a reaction-bonded mullite matrix with an SiO2 content of less than 1.5% is used. The use of this type of material prevents expansion of the channels and inclusions of alumina.

The article addresses important issues related to the development of a furnace lining material that would be more advantageous in terms of usability and strength. However, I have a few comments:

11. Usually pure quartzite is used as a cheaper suitable working material for furnace lining, but special-purpose lining already contains more resistant materials such as: quartzite / zirconium oxide (90/8%), quartzite doped with fused silica, andalusite, corundum. Is there a need, then, to improve the quartzite itself through costly and energy-consuming treatments? critobalite itself is formed at a temperature above 1000°C with a relatively long holding time at this temperature (even up to 5 hours). I do not think that this type of energy-consuming treatments could have a future if there are ready-made materials with better properties available on the market.

Fig. 1-3 – very poor quality of pictures.

3.       The studies are described extensively and meticulously.

4.       The abstract informs about the aim of the research, which is to be the effective production of foundry products for enterprises operating induction crucible furnaces with industrial frequency. But in your conclusions you only mentioned this possibility. Besides, how at 200°C degrees a cristobalite phase can appear, when this transformation is most fully and preferably in the temperature range from 1200°C to 1300°C, and the complete transformation into cristobalite (e.g. in laboratory conditions) takes place after 2-5 hours of the material being in furnace at a temperature not higher than 1350-1400°C? Does this admixture of cristobalite sometimes occur naturally in this quartzite?

Author Response

Dear Reviewer 3,

Thank you for your remarks on our paper, so we’ve made changes in the article.

Remark 1: Currently, the world is striving to implement the idea of the so-called Zero Waste, which is a response to environmental damage caused by the consumer's lifestyle. The increasing amount of waste generated and the consumption of natural resources too quickly is driving many people to "switch to Zero Waste". This idea does not bypass the foundry industry, where apart from recycling, for example, metal scrap, we are also dealing with the saving of natural resources (refreshing and reclamation process of moulding sands) and the production of ecological binders. Cast metal manufacturers rely on different types of furnace technologies. Although blast furnaces are often associated with metal working, they’re actually used for extracting iron and other metals from ores. Manufacturing foundries need other furnaces to take metal alloys and additives and make them into certain grades of cast metal. The most common four furnaces in use at a manufacturing foundry: induction furnaces; crucible furnaces; cupola furnaces; electric arc furnaces. The purpose of the lining is to prevent the furnace from overheating and wasting energy. This can be achieved by using dense bricks and mortars, insulating bricks, refractory concretes or ceramic fiber materials, usually in the form of modules. Materials with a low thermal capacity have the advantage that they heat up and cool down faster, while materials with a higher thermal mass are more durable and wear-resistant.

Reply: Thank you for the comment.

Remark 2: Companies offer an extremely wide range of materials for lining blast furnaces depending on the specific installation conditions and loads. Fired at temperatures up to 1700°C, corundum bricks are characterized by high resistance to abrasion and changes in thermal conditions, thanks to which they withstand mechanical stress in the most loaded, upper part of the furnace. In the reduction zones, special chamotte and andalusite fittings are used, which ensure exceptional resistance to the reducing environment. The high temperature and pressure generated by the charge put an enormous strain on the melting zone of the furnace. Ring elements made of chemically / ceramically bonded andalusites have been developed for highly stressed areas. The bottom of the furnace, the hearth, is covered with many layers of carbon brick. Thanks to the use of low-iron and low-iron chamotte grades and good heat resistance, which is combined with high-quality sintered mullite (with exceptional corrosion resistance), the "ceramic cup" of the furnace can be protected from the effects of pig iron. There is an increasing demand for innovative heat-resistant systems that increase the efficiency of entire plants. In addition to the high stresses generated by the charge, atmosphere and temperature variations, the lining is also negatively affected by the increased flow of materials, thereby causing an additional increase in the friction level. In modern foundries, the channels for ingot molds are lined with the highest quality refractory products. The produced steel must be free from non-metallic inclusions and have a stable composition and homogeneous structure. In the case of high-quality high-alloy steels, such as manganese steels, a material with a reaction-bonded mullite matrix with an SiO2 content of less than 1.5% is used. The use of this type of material prevents expansion of the channels and inclusions of alumina.

Reply: Thank you for the comment.

Remark 3: The article addresses important issues related to the development of a furnace lining material that would be more advantageous in terms of usability and strength. However, I have a few comments:

  1. Usually pure quartzite is used as a cheaper suitable working material for furnace lining, but special-purpose lining already contains more resistant materials such as: quartzite / zirconium oxide (90/8%), quartzite doped with fused silica, andalusite, corundum. Is there a need, then, to improve the quartzite itself through costly and energy-consuming treatments? critobalite itself is formed at a temperature above 1000°C with a relatively long holding time at this temperature (even up to 5 hours). I do not think that this type of energy-consuming treatments could have a future if there are ready-made materials with better properties available on the market.

Reply: Thank you for the comment. For industrial frequency induction crucible furnaces the following types of lining are used: acidic, basic and neutral, which are used either as a finished liner mixture or purchased materials from which, directly in the foundry made liner mixture (it is the cheapest). The practical resistance of the basic and neutral lining is 60-100 melts, the resistance of acid-300-350 melts. Price questions - Corundum mass 70 rubles per 1 kg, Spindle forming lining mass Coral CXL-500 rubles per 1 kg, Heat-resistant lining of induction furnaces for smelting alloy steel and cast iron at temperatures up to 1800 rubles per 180 kg, Mass dry sour quartzite Indastro Firebond MIX FS98 0.4 at a price of 14 rubles per 1 kg and quartzite ground PCMVI 2 at a price of 11 rubles per 1 kg.

Remark 4: Fig. 1-3 – very poor quality of pictures.

Reply: Thank you for the comment. The drawings are made with photos of equipment in the current technological process, so the clarity of contrast and brightness are not always optimal. But we have tried to corrected the quality of figures 1-3.

Remark 5: 3.       The studies are described extensively and meticulously.

  1. The abstract informs about the aim of the research, which is to be the effective production of foundry products for enterprises operating induction crucible furnaces with industrial frequency. But in your conclusions you only mentioned this possibility. Besides, how at 200°C degrees a cristobalite phase can appear, when this transformation is most fully and preferably in the temperature range from 1200°C to 1300°C, and the complete transformation into cristobalite (e.g. in laboratory conditions) takes place after 2-5 hours of the material being in furnace at a temperature not higher than 1350-1400°C? Does this admixture of cristobalite sometimes occur naturally in this quartzite?

Reply: Thank you for the comment. The proposed technology can be used only for industrial frequency induction smelting crucible furnaces. 200 °C is for the quartzite drying operation, which removes moisture, then it is cooled to perform the technology of preparation of the lining mixture. Next, the kiln crucible is packed with this mass and then sintered according to a schedule dependent on the kiln’s capacity. The sintering process ends at a temperature of 1550°C, resulting in a sintered layer.

With best regards,

Dr. Vadim Tynchenko

Round 2

Reviewer 1 Report

The revised version of the manuscript shows the significant impovement. I am very satisfied the revisions by the authors. I think it can be published in the present form.